# Perception of Recycled Plastics for Improved Consumer Acceptance through Self-Reported and Physiological Measures

**DOI:** 10.3390/s22239226

**Published:** 2022-11-27

**Authors:** Ainoa Abella, Pere Llorach-Massana, Alexandre Pereda-Baños, Lluís Marco-Almagro, Miguel Barreda-Ángeles, Laura Clèries

**Affiliations:** 1ELISAVA, Barcelona School of Design and Engineering, UVIC-UCC, La Rambla 30-32, 08002 Barcelona, Spain; 2Department of Statistics and Operations Research, Universitat Politècnica de Catalunya | BarcelonaTech, Avinguda Diagonal, 647, 08028 Barcelona, Spain; 3Eurecat, Centre Tecnològic de Catalunya, C/Bilbao, 72, 08018 Barcelona, Spain

**Keywords:** social perception, recycled materials, electrodermal activity, electromyography environmental attitude, consumption habits, sensorial properties, plastic product design

## Abstract

This article aims to provide in-depth insight into how consumers perceive recycled materials in comparison with natural raw materials at both the perceptual and attitudinal levels. To this end, we combined classic self-reported measures of sensory aspects, preferences, environmental attitudes, and consumption habits together with physiological measures of cognitive–emotional processing. Three different materials—two recycled materials, M2 and M3, and one raw material, M1—were chosen for inspection through three different sensory conditions, which we refer to as channels —visual, tactile, and visuo-tactile. The assignation of materials to sensory channels was counterbalanced so that each participant evaluated only one of the materials per channel. Although participants in general were not very accurate in discriminating between the materials, self-reported sensory evaluations showed that M3 (a recycled material that is made to look non-recycled), was clearly less liked. Meanwhile, the psychophysiological analyses revealed higher levels of electrodermal activity for the tactile evaluations of both recycled materials (M2 and M3). Finally, the results from the attitudes and habits evaluations indicate that the participants had positive environmental attitudes yet poor consumption habits. Altogether, these results suggest that some sensorial properties differ between recycled materials and natural raw materials and that there is a chance to improve and implement new consumption habits. The implications of these results are further discussed both in terms of suggestions for designers and methodological recommendations for researchers.

## 1. Introduction

Plastic production has increased twenty-fold over the last 40 years. Concretely, over 300 million tons of plastic are now consumed every year [1]. Out of all the plastic ever produced, by 2018, a mere 9% has been recycled, 12% has been incinerated, and the remaining 79% has gone to dumps, landfills, or nature [2]. Clearly, solutions to reduce the ecological impact of plastic consumption are greatly needed. An obvious solution is to reduce plastic waste by means of recycling.

The circular economy provides a set of approaches that could reduce plastic waste. International initiatives have arisen that promote plastic recycling within the context of the circular economy, such as “The New Plastics Economy” [3]. This initiative defines a series of actions that companies and institutions could apply to reduce plastic use and increase recycling. Along this line, the European Commission approved the “European Strategy for Plastics in a Circular Economy” published in 2018 [4]. This strategy banned single-use plastics by 2021 and established that plastic bottles must be produced with 25% recycled plastic by 2025 and 90% by 2029. Three years after the publication of this strategy, in July 2021, only five countries—Estonia, United Kingdom (despite now being outside of the EU), France, Greece, and Sweden—had already transposed the directive from the Plastic Strategy into national laws, whilst most countries were still preparing the laws, and less ambitious five countries had not started to apply effective measures [5].

Most of the strategies that these initiatives propose focus on the redesign of plastic products, highlighting the key role designers will play in this New Plastics Economy. For example, to help designers increase product circularity, including products made of plastic, the Ellen MacArthur Foundation has drafted a *Circular Design Guide* in which different circular design toolkits are provided. Delft University has also published the books *Products That Last* and *Products that flow* which provide different inputs for designers to introduce circular economy principles in the design process. In addition, designer Leyla Acaroglu has published different handbooks and toolkits for this purpose.

Some companies are starting to become aware of the pressing need for designing products made from recycled plastic. Businesses that are not yet aware will find themselves forced to produce with recycled plastic by new legislation sooner or later. One of the main concerns companies have is that products made with recycled plastic could be perceived by consumers as being inferior in quality, negatively affecting their sales and earnings. Therefore, studying how consumers perceive raw and recycled plastics is essential if we want to determine the best way to use recycled plastics in products and increase consumer acceptance. This paper addresses this question as its key element.

## 2. Background of the Research

Previous studies have analyzed the perception of single-use plastics [6] or the perception and opinion of plastic debris [7]. Other studies, which have focused on the perception of plastic materials applied in products, have studied how to improve the perception of bio-plastics in products from a design perspective [8] or the perception of products made from recycled ocean plastic [9]. In the last few years, especially during 2020 and 2021, new studies have been focused on exploring the perceptions and behaviors of different stakeholders. In order to evidence the previous statement, the keywords “recycled plastic perception” and “recycled plastic opinion” were used for searches with the help of several online databases—i.e., Scopus, ISI Web of Knowledge, Science Direct, and Google Scholar. Some papers can be found that focus on perceptions, acceptance, and attitudes regarding recycled materials.

Among the studies found, several research topics have been related to perception, especially of final products such as clothes [10,11] and electronic products [12], among others. On the other hand, there have been some first explorations on consumer perception and behavior [13,14] and the identity of recycled plastics [15,16].

Moreover, despite the abundance of literature on the perception of materials in general, there is also a lack of empirical research on the psychological (sensorial, cognitive, and emotional) aspects of the perception of recycled materials [15,17,18,19,20,21]. In the present study, the authors focused on the visual and tactile modalities, which are obviously the most relevant when it comes to the perception of materials. In this sense, it is a well-known fact to all sighted people that visually misjudging the physical properties of a material can often result in undesirable situations. However, it is also evident that this does not happen too often.

On the contrary, most human adults are quite proficient at handling these visual judgments despite the computational problems this entails for the visual system. There are also visual properties of objects that strongly appeal to touch, a fact that has been described in the literature as the “haptic invitation” effect [22]. Many studies have provided evidence of associations between visual and tactile sensations, showing, for example, a systematic matching between visual sensations such as lightness, black, and white, and vibrotactile ones such as low or high frequency [23]; systematic associations between smoothness, softness, roundness, and luminance [24]; associations of smoothness and softness with chroma (a measure of color intensity); and associations involving specific colors—for example, softness was associated with the color pink, while roughness was associated with brown. In view of this capacity to derive tactile information from the visual channel and vice versa, it is worth pondering the extent to which people are able to use this skill to detect whether a given material is recycled or not, with either of both sensory modalities. To this end, participants in this study experienced the target materials in three different sensory modalities, which we refer to as channels (see Section 3.3.2): visual only, tactile only, and visuo-tactile. The overarching question here is: to what extent does the appreciation of these materials depend on each of these sensory channels? Stated differently, should we aim to make recycled materials more appealing and should we focus on or ignore any of those channels?

This paper aims to better understand how consumers perceive recycled plastics through a heterogeneous methodological approach, including both self-reported and physiological measures, which, to our knowledge, is a novelty in this particular field. While self-reported methods provide information about the conscious perception of an emotion by the participant, psychophysiological methods can inform about cognitive or emotional processes beyond the participants’ awareness as they unfold, with high temporal resolution. Note that self-reported measures can be inherently flawed when it comes to describing such cognitive/emotional processes, as conscious awareness of the underlying psychological processes is very limited [25] and may also present problems related to different semantic interpretations or cognitive biases [26]. Since, to our knowledge, this methodology has seldom been employed in design studies, we believe this is a ripe opportunity to explore the potential of physiological measures in this research field.

The results will help us to define a list of strategies that designers and companies could apply to recycled plastic products to make sure that consumers do not perceive them as low-quality products.

## 3. Research Method

This study, framed as an experimental approach, aimed to discover how consumers perceive and discriminate recycled (R) and non-recycled (NR) materials, as well as their preferences, environmental attitudes, and consumption habits.

### 3.1. Participants—Sample

A total of 36 volunteer participants aged 20–67 years of age participated in the study. Participants were recruited through the networks of the research team and university students. In order to have a balanced representation covering both genders and young and old participants, a cluster sampling technique was used, dividing the population into men and women, and over and under the age of 35. A cutoff of 35 was used as the border between youth and adulthood [27]. Therefore, there were four groups of participants: women under 35, women over 35, men under 35, and men over 35. Each group consisted of 9 randomly selected individuals (see Figure 1). Participants had no remuneration for collaborating in the study.

Besides gender and age, the criteria for selecting participants were the following: participants had to be active consumers with varied professional profiles. The main difference between these four groups, apart from age and gender, was the information: INFO or NO INFO. In the INFO groups, all participants received a small amount of information making them aware that some materials were recycled and others were non-recycled, but without knowing which was which. The NO INFO group did not receive any particular information about the materials.

The sample in this psychophysiological study could be seen as small if it is compared to traditional research, but this is a sample size often used in this area, especially when the aim of the research is mainly exploratory (e.g., [28,29]). Moreover, larger sample sizes are hard to provide in this kind of study given the complexity of the process of collecting and analyzing data with highly time-consuming experimental protocols where researchers have to explain to participants the different techniques employed, place and remove sensors, etc. Moreover, materials and consumables are often costly as well, making the testing of larger samples rather prohibitive. This is a trade-off that is often assumed in exchange for a more complete understanding of the different aspects of human behavior, and its underlying causal factors, as mentioned above, are overt behavior, physiology, and subjective experience [30].

In any case, all conclusions extracted from the data analysis take into account the fact that it is based on a group of 36 participants. Statistical hypothesis testing, and in particular the study of the *p*-values and confidence intervals, was always used when analyzing the data. This fact guarantees—with a high level of confidence—that the observed differences are real and not due to randomness in the data collection process.

### 3.2. Procedure

The experiment took place in a small room with constant temperature and illumination conditions in order to ensure privacy and the same environmental conditions for all participants. Each experimental session began with the preparation of the physiological sensors and a relaxation phase lasting up to 20 min, after which the experimental session started. Each participant carried out three trials in the order determined by counterbalancing. Each trial began with the presentation of the corresponding material; participants were able to explore each material for 30 s, after that time the participants filled out the sensory evaluation test and proceeded to the next trial. At the end of the experimental series, participants filled out the environmental attitude and consumption habit questionnaires (see Figure 2).

### 3.3. Independent Variables

The independent variables in the study were the materials used—material M1, non-recycled; material M2, recycled that looks recycled; material M3, recycled that hides that it is recycled—and the channels used for the material inspection, with three levels—visual: V, tactile: T, and visuo-tactile: V+T.

#### 3.3.1. Materials

Three samples of plastic materials were used in the study. Each one of these samples is representative of the three kinds of materials we wanted to study: a non-recycled plastic, a recycled plastic that clearly looks recycled, and a recycled plastic that was created to be perceived as non-recycled. We could have used more than one sample of material for each group, but this was not possible due to economic and time restrictions (having each participant in the study evaluate more than three samples would require a lot of time).

The materials were put into cardboard boxes with identical dimensions—100 × 100 mm—and presented to the participants in the boxes. These squared samples were preferred since all consumable products could have a meaning and a function that could be directly related to emotional needs, therefore confounding the possible effects of the product representations. It would have been better to have all three materials with the same color, but this was not possible due to the availability of materials.

Each sample material was assigned an “M” nomenclature instead of its real name—M1, M2, and M3—to avoid giving any kind of information about the materials and to compile the data in the most neutral manner possible.

The materials used in the study (see Figure 3) are explained below:M1: The raw material sample. Red opaque polymethyl methacrylate (PMMA) foil, with the trade name Altuglas [31]. It is used in architectural and urban projects, visual communication, and within the health sector and automobile industry;M2: The recycled material with an evident recycled appearance. High-density polyethylene (HDPE) plastic carpentry made from tetra pak milk cartons [32];M3: Syntal [33], the trade name of the recycled material that cannot be easily identified. These are extruded profiles of recycled polyethylene and polypropylene, with an acrylic outer layer.

#### 3.3.2. Sensory Channels

In this study, we considered three different inspection conditions based on the sensory modalities used in each of them: visual (V), tactile (T), and visuo-tactile (V + T). In the visual channel, participants inspected the material using the sense of sight, without any other possibility of contact. In the touch channel, participants closed their eyes and could only actively touch the sample material. The visuo-tactile channel was multimodal and allowed participants to inspect the materials combining both channels, visual and tactile.

The procedure for the materials inspection was carried out in three rounds. Each participant inspected each of the three materials (M1, M2, and M3), using each of the sensory channels (V, T, V + T). For instance, one participant could use M2 in the first round with channel V, M3 in the second round with channel V + T, and M1 in the third round with channel T. In every round, both the sensory channel and the material were new for that participant.

Each user had a different pathway assigned so the presentation order of the materials and the channels were randomized. Figure 4 shows the possible combinations used for both the channels and materials. When combining the channel combinations (left column) with the material combinations (right column), a total of 36 different paths is obtained. As we had 36 participants in the study, each path was randomly assigned to one participant.

### 3.4. Dependent Variables

There were three self-reported dependent variables and two physiological variables. The self-reported variables came from three different questionnaires. The first questionnaire was administered after each trial and consisted of a sensory evaluation test of the submitted material sample (see Figure 5). The two other questionnaires concerned environmental attitudes (see Figure 6) and consumption habits (see Figure 7), and were both presented at the end of each experimental session. As the participants were Spanish and Catalan native speakers, in order to not affect the description by language properties, all three questionnaires and the first instructions for filling out them were in these two languages.

Although recordings were obtained throughout the whole test, we only analyzed the physiological data during the events of interest, namely, watching the material, touching it, or both.

#### 3.4.1. Physiological Measures

To evaluate the participants’ emotional experience, we adopted the dimensional model of emotion, which conceptualizes emotions as a function of two factors: hedonic valence, i.e., the degree to which emotion is positive or negative, and arousal, i.e., the intensity of the emotion. The dimensional model is more suited to quantitative analyses than the alternative categorical model, which focuses on the discrete labeling of different emotional experiences, such as surprise, happiness, and disgust. Two physiological measures—electrodermal activity (EDA) and facial electromyography (EMG)—on the corrugator supercilii muscle acted as proxy measures of arousal and valence, respectively. Specifically, EDA [34] refers to the conductivity of the skin, which varies depending on the activity of the sympathetic nervous system and is used as an index of arousal. Facial EMG measures the activity of facial muscles, which provides information about the valence of the emotions. Specifically, activity occurs over the corrugator supercilia, a muscle above the eyebrow [35,36].

The measurements of the EDA and EMG were collected using the Biopac MP-150 system and visualized and recorded using AcqKnowledge software, version 4.4.2.

The EDA data collection involved attaching two electrodes to the middle phalanx of each participant’s non-dominant hand [35], while the facial EMG data were collected by attaching two electrodes to each participant’s face [36]. The placement of the electrodes was conducted 15 min immediately before exposure to the materials began. The physiological measures were collected while the participants explored the three samples of materials (M1, M2, or M3) through a different channel (visual (V), tactile (T), or visuo-tactile (V + T)) according to the specific procedure/path assigned. After inspecting one material for 30 s using a specific channel, the participants evaluated the sample (M1, M2, or M3) using the sensory evaluation test (see Figure 5).

#### 3.4.2. Sensory Evaluation Test

The design of the sensory evaluation test was inspired by the Perception Evaluation Kit [37].

The test has three different parts: the first impression of the material, the evaluation of all the sensory properties (opposite pairs), and questions about the recycled nature of the material and possible applications.

The procedure was repeated for each material sample according to the corresponding channel. The sensory evaluation test was always the same. Some of the properties seemed difficult to evaluate in some channels (such as glossy–matte when only using the tactile channel). However, participants were asked to give a rating to all properties even if they found it difficult, as research on tactile–visual synesthesia has provided ample evidence of sensory associations between these channels. For instance, a person can infer a material is glossy when touching a slippery surface.

In order to facilitate subsequent analysis of the data, an icon in the upper-left part of the document indicated the interaction channel. The maximum time taken to fill out the information was about 4–5 min per file.

#### 3.4.3. Environmental Attitude

After participants were exposed to all three materials, the electrodes and other elements were disconnected and removed to allow the participants to feel more comfortable in order to answer the environmental attitude questionnaire (see Figure 6). The time needed to answer this questionnaire was approximately 5–7 min. Environmental attitude and participant consumption habits (see Section 3.4.4) were considered in the study to determine if a better environmental attitude and consumption habits were associated with a better capacity to differentiate recycled materials from non-recycled ones.

First, the questionnaire asked which material participants liked the most, which the least, and why. Second, a series of statements related to the planet’s current condition, plastics, animal rights, resources, the Earth’s capacity, and other related questions were presented. The series of statements were created by selecting some of the statements from the revised New Environmental Paradigm (NEP) [38], a revision of the NEP concept conceived in 1978. The following issues were considered from a sociological point of view: (1) our social beliefs about the potential for technology to save our world and (2) the idea that we live on a planet with infinite resources. These common social beliefs are called dominant social paradigms (DSPs). As observed, DSPs have led us to create a non-sustainable society. For this reason, Dunlap and KD Van Lier considered that a new, opposing social belief should be created to lead us to a more sustainable society, which they called the NEP.

The NEP is a list of 15 pro-environmental statements that can be used to determine people’s environmental consciousness by asking about their level of agreement with the statements. From this list of 15 statements, we selected the seven that we considered could be easily understood by participants to create a short and quick questionnaire. Additionally, two extra statements that focus on the perception of recycled plastics were added (see the last two statements on the questionnaire shown in Figure 6). The participants were asked to give a score from 1 to 5 depending on the level to which they agreed or disagreed with a statement. Subsequently, the affirmations were scored in a way that allowed for quantifying each participant’s environmental attitude. For the questions where the right end of the scale meant the maximum level of environmental consciousness, the ratings went from 1 to 5. For the questions where the left end of the scale meant the maximum level of environmental consciousness, the ratings went from 5 to 1. The ratings of all questions were summed up, and the final rating was scaled to obtain a final figure going from 0 to 100 (0 meaning a person with no environmental consciousness at all, and 100 a person with the maximum level of environmental consciousness possible). Open-ended questions 1 and 2 were studied to obtain descriptive information, but no relevant conclusions were extracted.

#### 3.4.4. Consumption Habits

The objective of this phase was to understand and analyze the consumption habits of participants and, afterwards, to find out if there was any correlation between their environmental attitude and their consumption habits. Participants were given a questionnaire to complete (see Figure 7) that consisted of four phases: criteria used to select purchases, questions related to packaging, recycling, and reuse, the participant’s recycling habits, and, finally, consumption in view of all this information. The average time dedicated to filling out the questionnaire was about 5–10 min.

The participants’ answers were scored in a way that facilitated quantifying each participant’s consumption habits. The score was obtained by adding a rating from 1 to 5 to questions 4.2, 4.3, 4.5, 4.6, and 4.7. Later, the figure was scaled to obtain a number going from 0 to 100 (0: worst consumption behavior, 100: best consumption behavior), as we did with the environmental attitude questionnaire. Questions that did not require a rating were studied in a qualitative way.

## 4. Results

The most suitable statistical methods for analyzing the collected data were used in the study. In addition to Excel for data recompilation and organization, we used Minitab (Minitab Inc.—Viena, Austria, 2020) and R (R Core Team—Coventry, United Kingdom, 2020) as statistical software packages for analysis. The accuracy of the identification of a material as recycled or not-recycled was computed based on the number of times the participants in the study correctly identified the material. Chi-Square tests have been used to evaluate if there are significant differences in this accuracy. For the analysis of physiological measures (EDA and EMG), mixed multilevel models were applied. We used the EDA and EMG continuous measurements as responses for these models: material, channel of presentation, and their interaction as fixed effects, and the main effect of the person evaluating as a random effect. Having this random intercept allowed for taking into consideration the possible different basal levels of response in each person, thus extracting this variability among participants and improving the ability of the model to detect differences among materials and channels. For the characterization of the materials based on the sensory evaluation data, we first conducted an exploratory cluster analysis. The aim was to be able to group sensory word pairs so that the later interpretation of the results would be clearer.

### 4.1. Sensorial Evaluation of the Materials

A cross tabulation with a Chi-Square test showed that there are no differences in the identification accuracy (see Figure 8 and Table 1), with high *p*-values for all presentation modes. When breaking down the data from all presentation modes, a new Chi-Square test showed that no differences in the identification accuracy could be detected among the three materials (Table 2).

The EDA was measured during the materials inspection in micro-Siemens (µS), processed prior to analysis following standard procedures, and averaged for each participant and condition. Mixed multilevel models were applied for the analysis. The models included, as fixed predictors, the mode of presentation, the material, and their interaction. For the mode of presentation variable, the visual condition was taken as the baseline condition, against which the rest of the conditions were compared. For the material variables, the baseline condition was Material 1. A random intercept was included that accounted for differences in the individual baseline levels, which is key in the analysis of psychophysiological measures [39].

Table 3 shows that the material and channel interaction was significant at a 5% level of significance, thus making both factors, the material and channel, active when using the EDA response. We used graphical outputs to facilitate the interpretation of the model.

Figure 9 shows the EDA level depending on the material, having removed the participant’s random effect. Material 3 elicited a higher EDA response, suggesting a higher sympathetic activation for that material as compared to Materials 1 and 2.

The model included statistically significant interactions between the materials and the presentation channel (see Table 1). Figure 10 shows a different material profile depending on the channel. The visual and visuo-tactile channels had high EDA levels for Material 3 and a low EDA level for Material 1. Conversely, the tactile channel shows the opposite pattern: high EDA for Material 1 and low for Material 3.

The same mixed-model approach was used with the EMG instead of the EDA as a response, using the same independent variables. In this case, no significant effects or interactions were observed for the EMG data, as Table 4 shows.

### 4.2. Characterization of the Materials

This section shows how the participants perceived the three materials based on the sensory evaluation questionnaire they filled out for each material to which they responded. Therefore, the data used were self-reported on a questionnaire. A hierarchical cluster analysis was performed in order to group the perceptions and facilitate interpretation (see Figure 11). Four different groups were created based on the dendrogram, “cutting” the dendrogram where the distance values among groups showed a high change from the previous step. A k-means clustering algorithm was later conducted with the data, using the previously identified four groups as an initial partition. The k-means algorithm does not show differences in the grouping. The fact that the four groups can be characterized without great difficulty also shows that the partition is adequate.

Figure 12, Figure 13, Figure 14 and Figure 15 summarize the results of the sensory evaluation according to the cluster grouping. The graphs show, for each of the sensorial properties, the average scores given for each material (M1, M2, and M3) and for each presentation channel (V, T, and V + T).

The properties listed above (Figure 12) show similarities between M2 and M3, as they were perceived as non-reflective, matte, and opaque. Conversely, M1 stands out with respect to the others. The tactile channel (T) provided less information compared to the other two channels regarding these properties because they are related to the sense of sight.

According to Figure 13, all materials were scored as colorless and cold regardless of the interaction channel. The light–heavy pair remained at a neutral value.

The graphs presented above (Figure 14) show the fact that all materials, regardless of whether they were recycled, were perceived as smooth and artificial, with no significant differences between the channels. The property related to color intensity remained near the neutral area.

Finally, there were four properties related to hardness and rigidity. All materials were overwhelmingly perceived as hard, tough, non-elastic, and strong (see Figure 15). The tactile channel (T) and visuo-tactile channel (V+T) yielded very similar scores for those properties.

### 4.3. Environmental Attitude and Consumption Habits

The following were considered from a sociological point of view: (1) our social beliefs about the potential of technology to save our world and (2) the idea that we live on a planet with infinite resources.

A scatterplot of the scores for consumption habits (see Section 3.4.4) versus the scores for environmental attitude (see Section 3.4.3) shows no correlation between these two variables (see Figure 16).

The coefficient of correlation was 0.029, with a 95% confidence interval (−0.303–0.354) clearly containing the value of zero. This result is the same when stratifying by gender or age group (with coefficients of correlation always including 0 for a 95% confidence interval.). The correlation between consumption habits and the percentage of correct identifications of the material was non-significant (95% confidence interval for the coefficient of correlation: (−0.016–0.58)). The correlation between environmental attitude and the percentage of correct identifications of material was also non-significant (95% confidence interval for the coefficient of correlation: (−0.33–0.33)). Therefore, we can conclude that environmental attitude and consumption habits did not affect the accuracy of identifying the materials.

## 5. Discussion

This study aimed to advance our understanding of consumer perception, acceptance, and attitudes towards recycled and raw plastics, with a novel methodological approach combining techniques from design research and experimental psychology. The main contributions of the paper are highlighted in Section 5.1. Section 5.2 proposes suggestions for designers to apply to recycled materials based on the results of this study.

### 5.1. Consumer Perception of Raw and Recycled Plastic

Regarding the sensory evaluation, the overarching question was whether the participants were more proficient at identifying recycled materials in unimodal (visual or tactile) or bimodal conditions, with a view to advising designers and retailers on how to better present these materials. However, although there were tendencies suggesting that participants were more accurate at identifying materials through the visual channel, they also needed to touch them to verify the nature of recycled materials. Regarding the hedonic appreciation of the materials, M1 and M2 appeared in the interval of positive results, which means that people liked them. On the contrary, M3 was on the negative side or within the so-called dislike area. Hence, recycled materials that do not show a granulate aspect need more time or concentration to be detected and can create ambiguity for participants. This could be observed in the hedonic results where participants penalized them. Regarding the physiological data, no significant effects were observed for the EMG, probably due to the materials not differing enough in the valence dimension, despite participants indicating less liking for M3. However, the EDA data revealed statistically significant interactions between the materials and mode, specifically a higher EDA activity was observed for Material 3 (the recycled material that cannot be easily identified), suggesting that the incongruence between the look and feel of the material could play a role in their dislike for the material. The tactile channel also showed higher EDA activity than the channels that incorporated vision. This could show that vision is a relevant sense in material recognition. Since it was quite challenging to observe differences when responding to the three samples, physiological measures probably needed more extreme or different stimuli to detect and understand people’s reactions. In any case, it is interesting to note that such an effect was not observed in the self-reported data, suggesting that this difference was not consciously perceived by the participants. As mentioned above, the lack of self-awareness of psychological processes is a very common phenomenon [25].

Therefore, looking at the combined results of the self-reported and physiological analyses, when it comes to sensory evaluation, the participants had more difficulties differentiating the recycled materials since they needed more information to classify them. This information could be given through the sensory properties of the material samples. In this paper, for which three samples were evaluated, the results suggest that natural raw materials, compared to the two recycled materials, were always perceived as more reflective, glossy, and colorful. Stated otherwise, recycled materials, in general, do not tend to transmit these bright properties because they usually emphasize matte properties and soft and pastel colors, even though is it technically possible. Regarding the other sensorial properties, all materials presented similar properties. In other words, the recycled materials looked like they could be perceived as having the same quality as other materials.

Although one could think that individuals with greater environmental awareness are more prone to have ecologically responsible consumption habits, such as recycling and reusing packaging, the self-reported data do not show this. This lack of a relationship between environmental awareness and ecologically responsible consumption habits shows that established habits are powerful and difficult to change and that awareness campaigns are probably not sufficient to change these habits. Other measures, such as a ban on using plastic bags could be more effective.

Moreover, the perception of differences between raw and recycled materials was also independent of environmental consciousness, showing that people assessed the materials in a multidimensional framework in which environmental consciousness was, at best, one of the dimensions. One could say that people within this group did not have a clear model regarding environmental attitudes because, as the results show, there is inconsistency in their responses.

### 5.2. Design Suggestions for Designers and Companies

In Section 4.1. (Sensorial Evaluation of Materials), we can observe that the perception and acceptance of raw plastics (sample M1) and recycled plastics (sample M2) were better than for sample M3 (recycled plastic that does not seem recycled). Two main aspects can be highlighted from these results. First, sample M3 seemed to create uncertainty for consumers, as it was not easy to identify if it was a recycled plastic or not, which made consumers feel uncomfortable. This situation led to a worse perception of sample M3. We could thus suggest that designers and companies should avoid attempts at hiding the fact that a material is recycled as consumers may perceive that brands are not being honest with them. Second, we should highlight that samples M1 (raw plastic) and M2 (recycled material) were similarly positively perceived by consumers. Given this result, we could infer that the myth that consumers do not like recycled materials because of their inferior properties, which is one of the main fears companies face when using recycled plastics, is unfounded. However, the increase in consumers’ environmental awareness during recent years could explain why consumers now perceive recycled plastics as a positive value for products instead of something that reduces product quality. In the case of Catalonia, the region where consumers were interviewed for this study, different initiatives have been created to increase environmental awareness over recent years. For example, the Catalan Waste Agency developed a series of awareness activities such as launching the Ecodesign Awards and producing new and more effective communication campaigns. Many of these awareness activities have focused on the positive effects of recycling on the environment, which has likely helped to increase consumer positive attitudes toward recycled materials. Consequently, if this new perception paradigm could be confirmed, companies and designers might not be reticent to use recycled plastics for products because recycled plastics now seem to be perceived as having a positive value when people decide on their purchases.

Another recommendation for designers and companies related to sensorial properties is the use of tools, such as a Perception Evaluation Kit, to evaluate user perceptions of a product or material sample before launching it on the market. This will provide them with valuable information on how to improve their designs and make them better perceived by consumers.

The results of this paper are related to the historical analysis of the evolution of material and product sustainability approaches regarding consumer products. In the nascent sustainability megatrend, now a regular feature in many consumer decision processes, products that claimed to be sustainable had intrinsic characteristics linked to sensory properties. Therefore, soft and matte material attributes linked to sustainable products have already become part of consumer culture, as can be seen in different trend reports. These data are in concordance with this paper’s results, which show that participants had a better hedonistic appreciation of materials that displayed more matte surfaces.

An evolution of the sustainability approach for consumer products arrived when composite waste-derived materials, normally plastic waste, were introduced. These recycled materials have the appearance of composite materials, particles that are joined or melted together to achieve a functional material or product with few manufacturing processes, such as *Smile Plastics* materials and *Freitag* recycled bags. It is important to reach the conclusion that in both cases (raw nature products or aggregate-type recycled sustainable products), the material or the product informs about the process through colors and textures.

The latest evolution of recycled materials has come with a more complex engineering process of recycling plastic waste, such as the transformation of plastic fishing nets into fibers and, later, fabrics such as *Ecoalf* products. The resulting material does not make the recycling process evident, employing flat surfaces and non-naturally inspired colors, and it is not different from other natural raw materials. Therefore, it is difficult to know if the material is recycled. However, these products are then put onto the market with great labeling, communication, and marketing campaign efforts, explaining where the product came from. If the material itself does not provide enough information through its sensory properties, the communication surrounding the product must take on the role.

Therefore, we can suggest that designers and companies either use recycled materials that are inspired by nature or have aggregate attributes. Alternatively, if the materials do not display these characteristics, they could support the product with educational and informative labeling and a marketing campaign.

## 6. Conclusions

Materials are an important aspect of our daily sensory experiences. Keeping in mind the actual plastic consumption and the possibilities of the circular economy, initiatives to understand how recycled materials can be better introduced to increase acceptance by users constitute valuable contributions. The results of this research help show the recent tendency of consumers to no longer perceive recycled materials in a negative way, as was often the case in the past, meaning that their uses can now be applied in other contexts. This indicates new opportunities and possibilities for the plastic industry and its different consumable products, such as furniture, clothes, packaging, and products associated with the food industry. That said, users have shown they are not very skilled at spotting differences between raw and recycled materials. This implies that designers must clarify the nature of recycled materials through the composition of the material or communication. Moreover, since users are not very adept at detecting materials, but their environmental attitudes and knowledge about the environmental situation are decent overall, there is great potential to improve and implement initiatives to correct their consumption habits which, for the moment, are not as good as their environmental attitudes.

Due to the similar nature of the materials chosen, the signals detected by the participants of this study were not extremely different from each other. The selection of materials showing better-expressed differences would probably have ensured more easily detectable differences between the emotional responses of the participants of the experiment. Additionally, the dissociation found between the subjective and objective measures of emotional responses is considered a common phenomenon [40,41]. While autonomic physiology reflects a fast biological non-conscious response to the stimuli [40], self-reported assessment requires conscious cognitive monitoring of the event that involves self-awareness and other relevant cognitive processes [41].

Moving forward, it would be interesting to carry out new case studies with a simpler experimental design and materials that will provoke more diversified responses. The use of other physiological measures could also help us to understand, step by step, which designs and materials add the most value to these kinds of experiments. Additionally, it would be advisable to focus on material research to understand which sensory properties make participants perceive materials as raw or recycled. With all these improvements and trials, the knowledge obtained thanks to such research can be used to create toolkits for designers to transmit gained information and to take the research to other areas, such as bioplastics, with the goal of creating valuable resources and information that designers can take into account during the design process.

## 7. Limitations and Future Research

Due to the different factors used and the nature of such a methodology, the final number of participants in each experimental group was low, reducing the strength of the experiment, but not the analysis or the main conclusions. In the future, using either more participants, a larger number of materials, and/or fewer stratification variables could be beneficial.

We believe this is a ripe opportunity to explore the potential of physiological measures in this context since this methodology has seldom been employed in design studies [42].

## Figures and Tables

**Figure 1 sensors-22-09226-f001:**
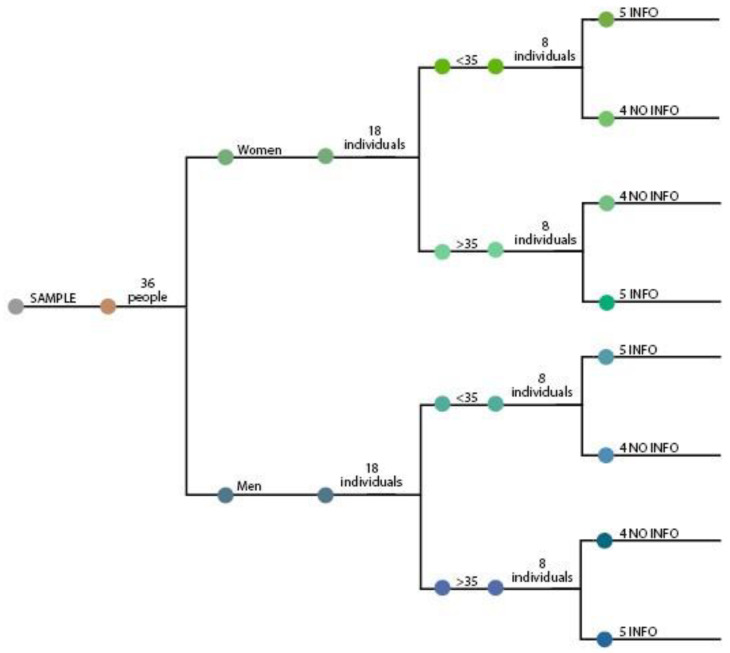
Sample distribution. (For further explanations see the text.)

**Figure 2 sensors-22-09226-f002:**
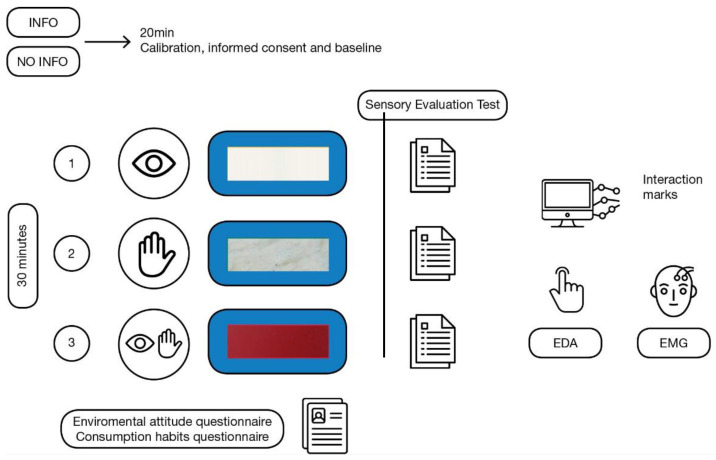
Procedure with EDA—electrodermal activity, EMG—electromyography, and sensory evaluation test.

**Figure 3 sensors-22-09226-f003:**
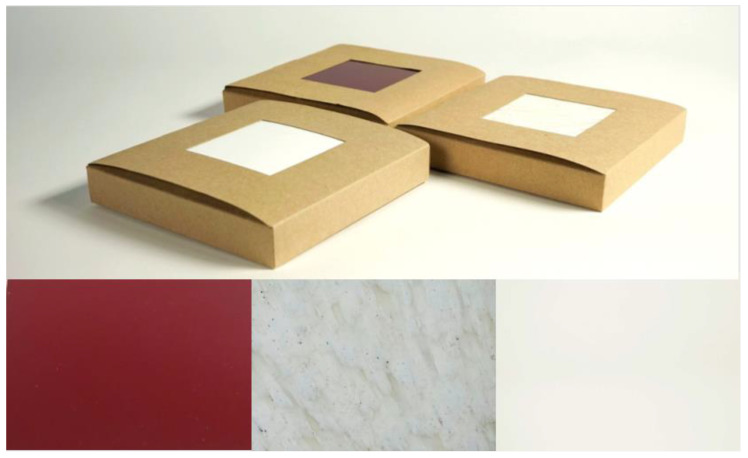
Material samples. In order from left to right: M1, raw material called Altuglas; M2, recycled material from Carpintería Plástica; and M3, recycled material called Syntal.

**Figure 4 sensors-22-09226-f004:**
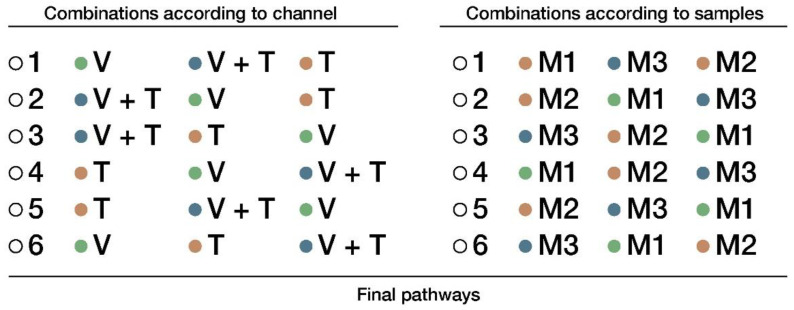
Combinations for material inspection through different final pathways.

**Figure 5 sensors-22-09226-f005:**
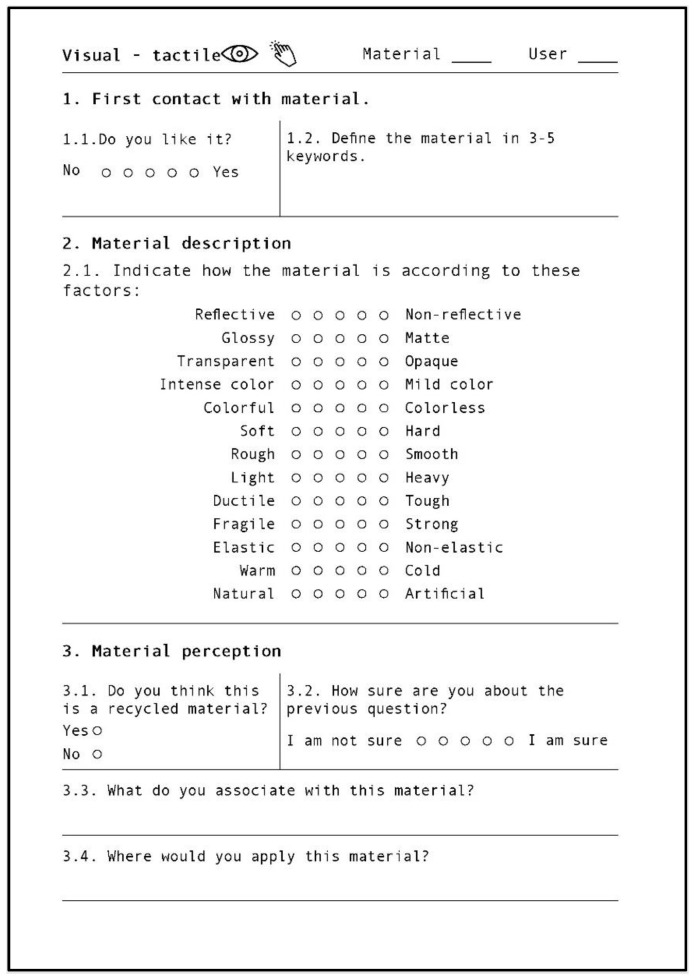
Sensory evaluation test.

**Figure 6 sensors-22-09226-f006:**
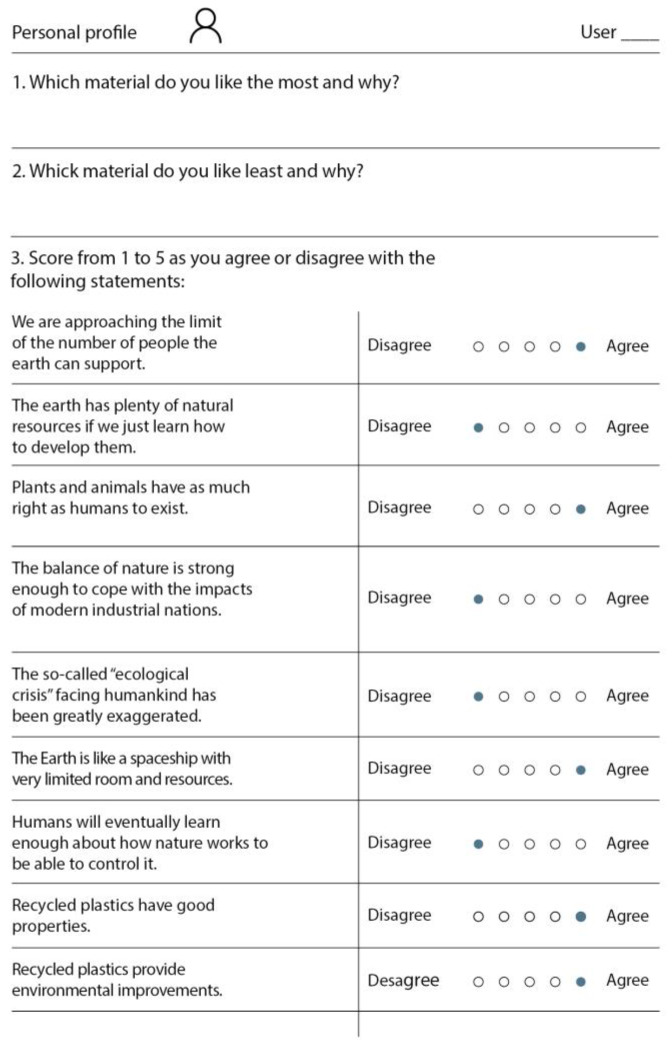
Environmental attitude questionnaire. A blue circle is used to show the answer that is associated with a better environmental attitude.

**Figure 7 sensors-22-09226-f007:**
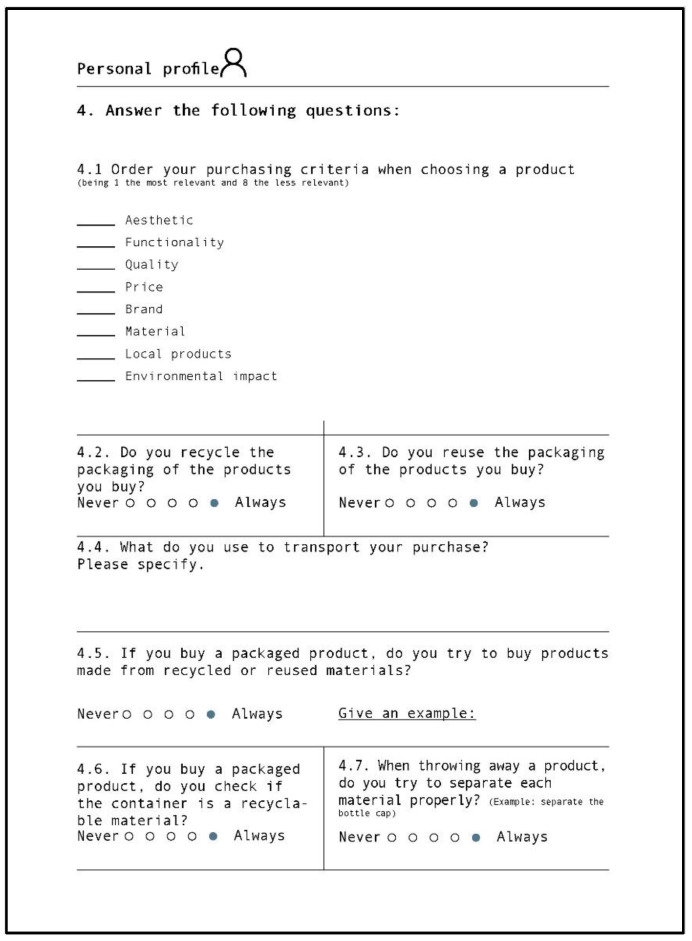
Consumption habits questionnaire. Blue circles show the answers that are associated with better consumption habits.

**Figure 8 sensors-22-09226-f008:**
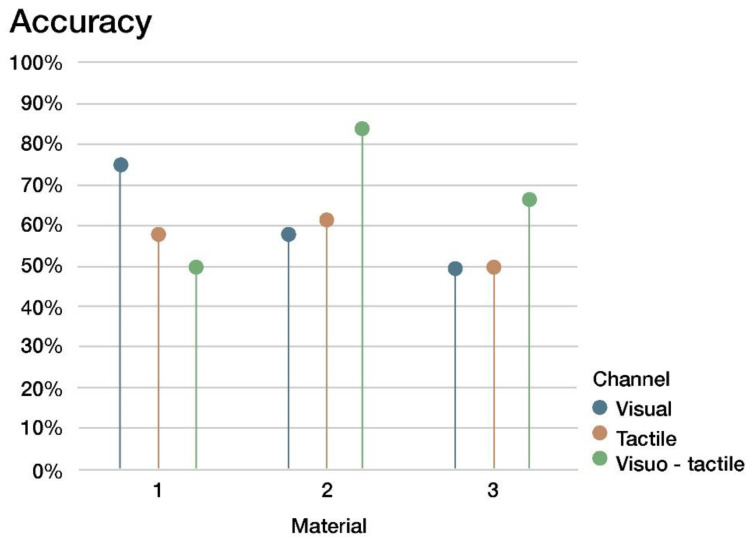
Accuracy in identifying materials when using different sensory channels.

**Figure 9 sensors-22-09226-f009:**
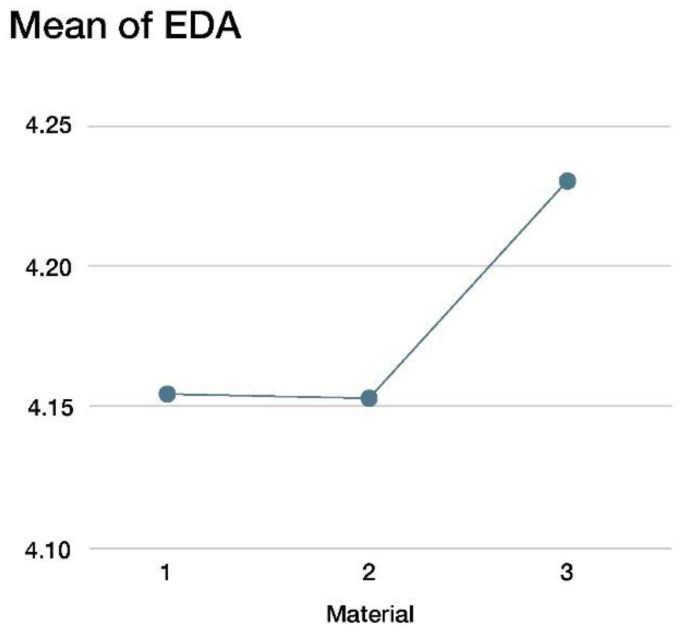
EDA response to various materials used in this study.

**Figure 10 sensors-22-09226-f010:**
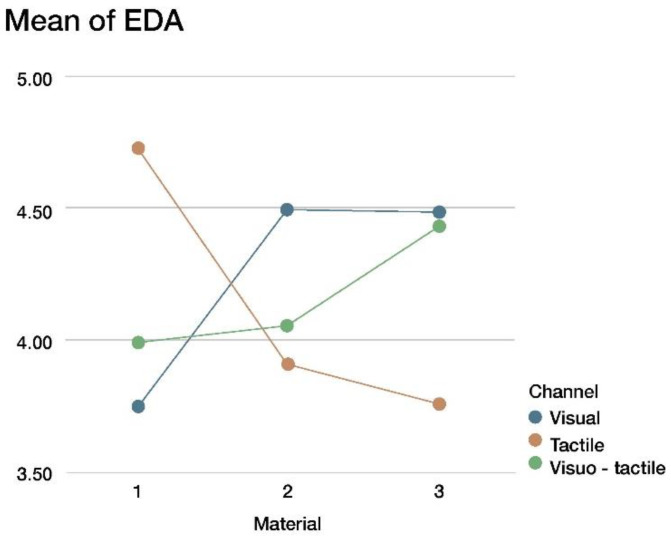
EDA response based on material and presentation channel.

**Figure 11 sensors-22-09226-f011:**
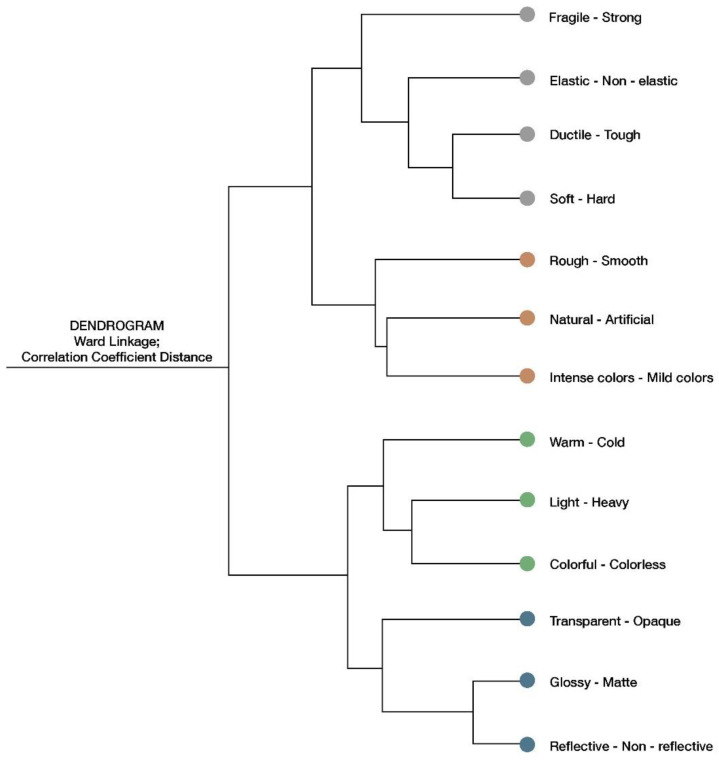
A hierarchical cluster analysis of the sensory descriptors performed using the Ward linkage method.

**Figure 12 sensors-22-09226-f012:**
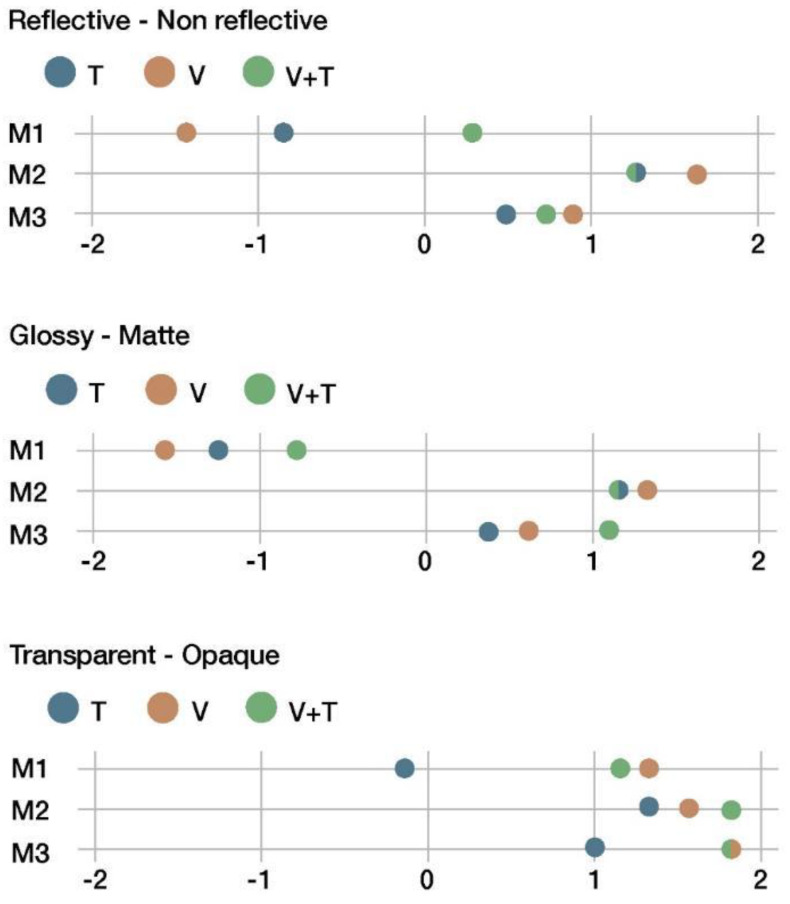
First cluster of sensorial properties.

**Figure 13 sensors-22-09226-f013:**
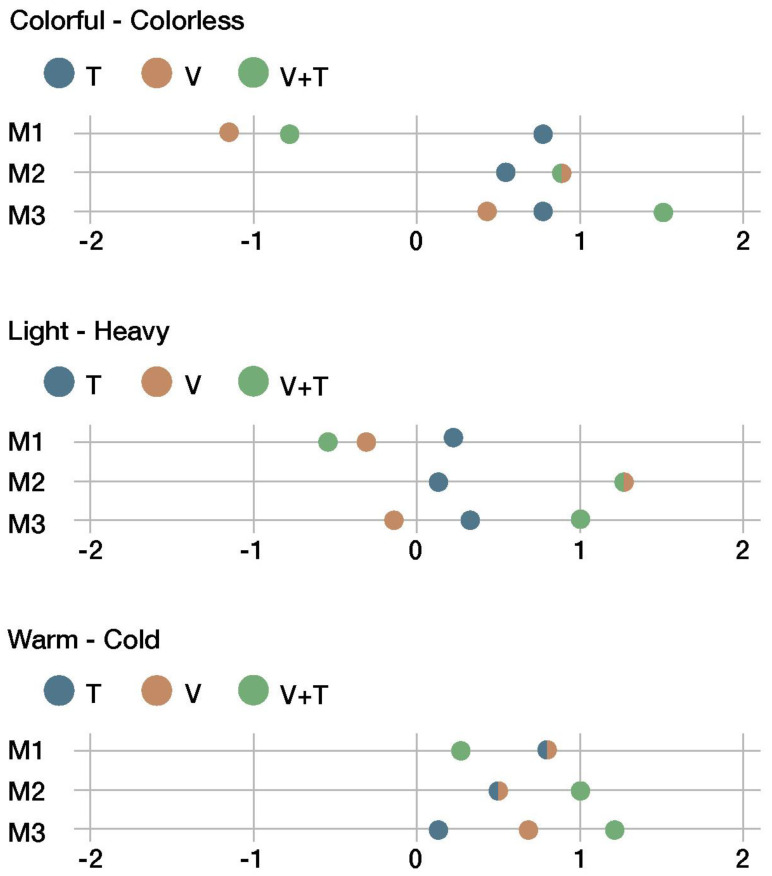
Second cluster of sensorial properties.

**Figure 14 sensors-22-09226-f014:**
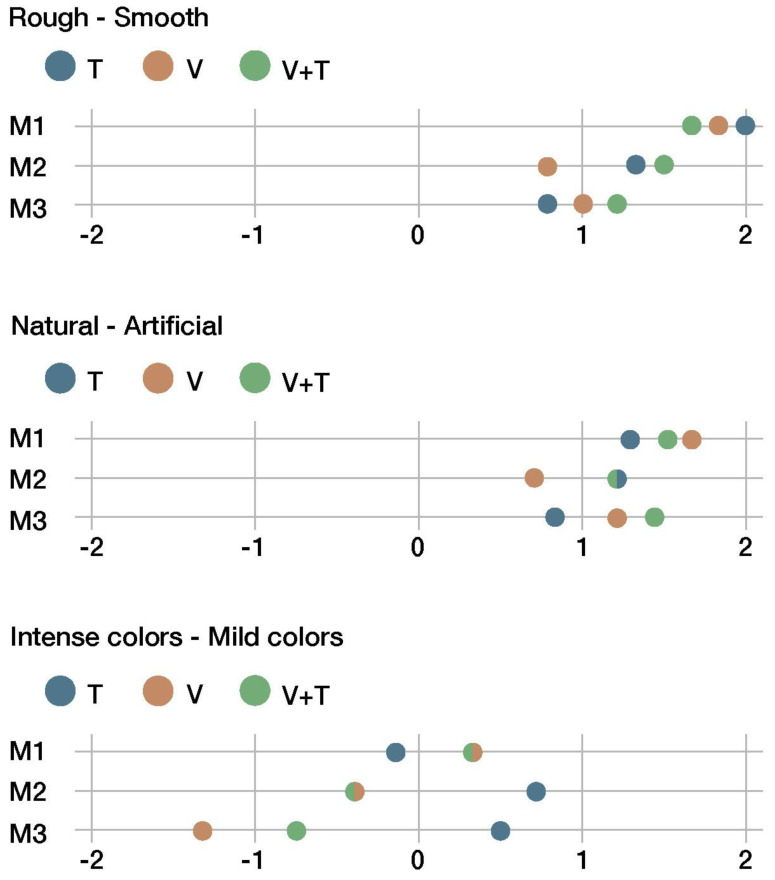
Third cluster of sensorial properties.

**Figure 15 sensors-22-09226-f015:**
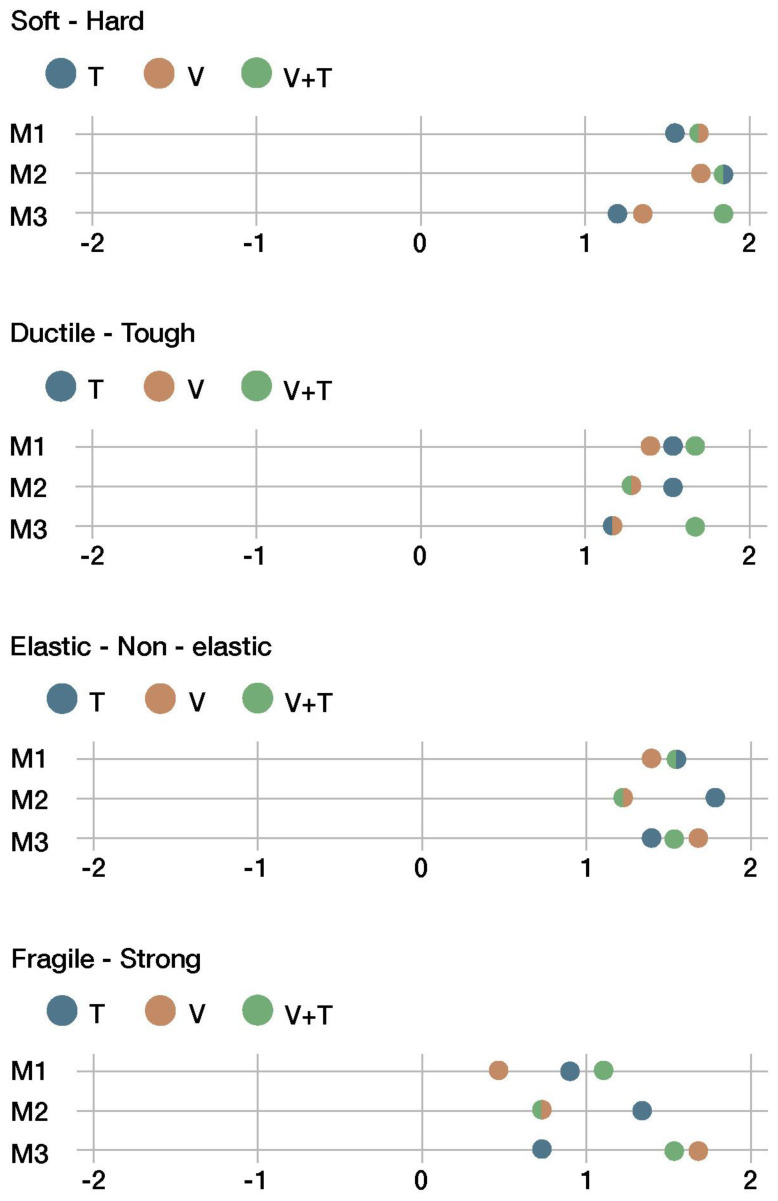
Fourth cluster of sensorial properties.

**Figure 16 sensors-22-09226-f016:**
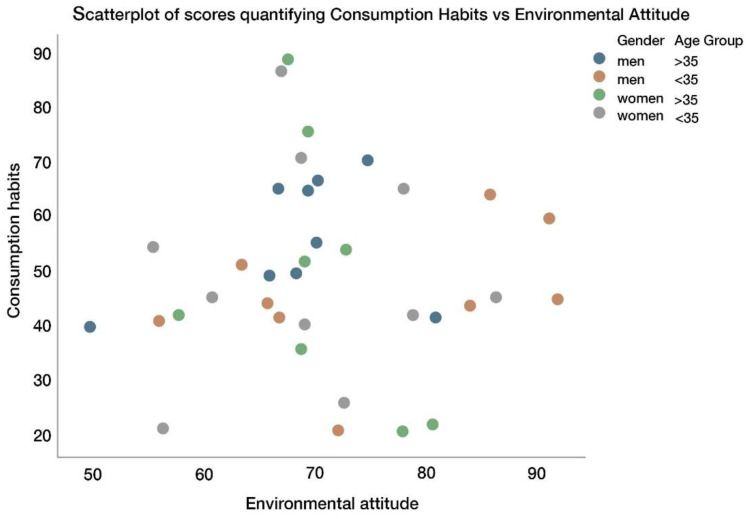
Scatterplot of scores quantifying consumption habits and environmental attitudes.

**Table 1 sensors-22-09226-t001:** Identification accuracy for each material and each presentation mode, with the results of a Chi-Square test for detecting statistically significant differences among presentation modes.

	Material 1	Material 2	Material 3
**Tactile**	58.3%	61.5%	50.0%
**Visual**	75.0%	58.3%	50.0%
**Visuo-tactile**	50.0%	83.3%	66.7%
***p*-value**	0.441	0.360	0.638

**Table 2 sensors-22-09226-t002:** Identification accuracy for each material, with the results of a Chi-Square test for detecting statistically significant differences among materials.

Material	%
**Material 1**	61.1%
**Material 2**	67.6%
**Material 3**	55.6%
***p*-value**	0.441

**Table 3 sensors-22-09226-t003:** Test of fixed effects in the mixed multilevel model using EDA as a response.

Term	*p*-Value
**Material**	0.523
**Channel**	0.824
**Material × channel**	0.026

**Table 4 sensors-22-09226-t004:** Test of fixed effects in the mixed multilevel model using EMG as a response.

Term	*p*-Value
**Material**	0.456
**Channel**	0.391
**Material × channel**	0.509

## Data Availability

The datasets generated during and/or analysed during the current study are available from the corresponding author on reasonable request.

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
