# Peer review of "Perception of Recycled Plastics for Improved Consumer Acceptance through Self-Reported and Physiological Measures"

_sensors, 2022, doi:10.3390/s22239226_

Round 1

Reviewer 1 Report

The article is highly relevant scientifically and practically (commercially), and it addresses a very important issue concerning the acceptance of recycled plastics by the end users of products. The strength of the article is its novel and unique experimental design. The findings are relevant and significant in the literature, as they fill a gap in knowledge about the general perception of consumers on the cognitive response to recycled materials. There are quite a few weak points in this article, especially considering the writing and the structure of the article, which I am indicating in the comments below.

Overall, this article presents significant findings and therefore I would gladly recommend it for publications, but with some major amendments. Here is a list of comments to consider for revising the manuscript:

1. A general comment for the manuscript is the need of proofreading. There are several mistakes here and there, typically with some syntax and vocabulary errors throughout the text, so a final version of the article would benefit from a thorough proofreading. 

2. The abstract needs rewriting. Typically, the abstract is a single paragraph continuous text, not split in two. Please provide a simple and conceptually clear abstract that is easy to read, so that the reader can have a quick impression about the content of the article.

3. The description of the materials “M2 from Carpinteria Plástica and M3, called Syntal, and one raw material M1 called Altuglas” is not needed in the abstract. Please consider removing.

4. Line 14, what is a holistic view? Better avoid such vague terms and write something more “to the point” of the matter, e.g. “an in-depth insight” etc… Please rephrase accordingly.

5. Line 15, the term “raw materials” is also not specific enough. Recycled plastic is also a “raw” material for the manufacture of products. You could either use “natural raw material” or “virgin raw material”. The former sounds better in my opinion. Please make sure you keep it consistent throughout the manuscript.

6. Line 41, the sentence does not make sense, please reformulate.

7. Line 46, please refer the year that the strategy was published, it is 2018.

8. Line 50, the correct terminology is “transpose” a directive into national law, not “transform”.

9. Lines 55-56, are there any examples you know of design guidelines for circularity? It would be a good improvement for the article to name a few, other than EMF.

10. Lines 73-73 are not appropriate to be presented in this section. In the literature review section, you may present the findings of previous literature, but not present the way (method) in which you searched for the relevant literature. Consider removing or make a dedicated sub-section in the method section of the article.

11. Lines 85-87, please revise the sentence. It does not make sense; it is hard to understand what the authors mean.

12. Section 2 literature review would be better renamed something like “background of the research” or similar. In this section the authors present more information than literature review, including the reasoning of choices for the research and the research objectives. Therefore, the title “literature review” is not fitting.

13. Section 2 could be restructured as follows: General title “background of the research” or similar. Then split this section in three sub-sections. First, one sub-section about the research gap, where you present past literature and gaps you have identified, both in the subject and the methodology. Second, a sub-section where you present the aim of the research, what you want to achieve by the research in this specific article. And last, another sub-section where you present the objectives of the article, including how you expect to fill the research gaps and to try out a methodological approach that has not been used before. This way the text is much better structured and tied together to form a solid background section. Please consider revising this section of the manuscript.

14. In the methodology of the experiment, you used three materials but with different colours. Why is that? Can you explain this decision? You have presented in the literature background that different colours could have a different cognitive effect on people, without tactile exposure. Then why different colours of materials.

15. There is no sufficient explanation as to why research the environmental attitudes and consumption habits, especially when the experiment is about the properties of recycled materials and the acceptance of consumers. Please give a thorough justification for this decision in your research approach.

16. In lines 361-76 you are describing the method for analysing the data. This, too, is part of the method section and should not be included in the results section. Here you start the section straight away by presenting the (statistically processed) results.

17. Is there a section 4.2 in the manuscript? Did not find it.

18. What does the scatterplot mean (Figure 16). You could have developed the analysis a bit further to give a better insight of the data to the reader. Little can be understood now from the state of the graph.

19. Lines 498-99, this section should not present the results, but the discussion of the results. Please revise the text. You are now in section 5 – discussion, which means that you have already presented the results of the research in section 4, and in this section you discuss the implications of the results in design (practitioners) and literature (academics).

20. Lines 538-44, it is unclear what is the relevance of this result to the overall experiment design. In the next paragraph, lines 545-50, there is a better connection to the research. But this paragraph (538-44) is largely unjustified and does not add anything significant to the article. On the contrary, some of the suggested policy solutions are not substantiated and seem to be just simple normative arguments by the authors, irrespective of the literature on the matter. Is there a better way of integrating the results of this paragraph to the main objectives of the article? If not, then it would be better to remove this completely.

21. Line 552, it is not section 3.1. Please refer to the right section in the manuscript.

22. Lines 580-90, you must use relevant references. In this text you refer to advances in literature and historically, but you do not support this with the respective references. Please add references that support your statements.

23. Lines 591-96 do not make much sense. Please revise the text so it becomes easier to understand.

24. Line 616, the statement “the results of this research show that a vast group of consumers…” is an exaggeration. It would be better to tone down this statement and accurately report the findings of your research. The results showed an indication about your statement, only.

Author Response

We sincerely appreciate the reviewer’s comments and suggestions. We have considered all aspects mentioned by him/her. Please, find attached below our detailed response to the reviewer paying special attention to modifications done for each point mentioned. The response is done specifically for each comment as you suggested. 

Yours sincerely.

Reviewer 2 Report

This paper reports the results of a study carried out to investigate an interesting and important question: how human subjects perceive the recycled and non-recycled materials via different sensory channels, and what are their physiological responses during the inspection of such materials. The authors hoped that the answers to these questions might shed an important light on our knowledge concerning the  preferences of the consumers regarding recycled versus non-recycled nature of various products, and that they might also help the designers to promote recycled materials. However, the reviewed manuscript has such a number of weak points that, in my opinion, it needs a major revision. Unfortunately, analysis of recordings of two psychophysiological measures, electrodermal activity (EDA), and facial electromyography (EMG) did not yield particularly interesting results. The use of only one non-recycled material and only two recycled ones may also be considered to represent a weak point of this study. The decision of the authors to ask the tested persons to assess colours and other features perceived via the visual channel when inspecting a given material only via tactile channel is controversial, too. Quantification of experimental data and methods of their statistical analysis are not sufficiently well described. Some conclusions are not supported by experimental data. The order of presentation of informations is not always optimal and, therefore, the text of the manuscript is rather difficult to follow. For instance, terms such as "corrugator supercilii" should be explained at their first appearance in the text, and if the information necessary to understand a given statement and/or figure is provided further in the text, this should be indicated. The text also contains repetitions of exactly the same text fragments and numerous other linguistic shortcomings that in many cases do not allow the reader to grasp the precise meaning of specific statements. I put detailed comments (in total, 136) within pop-out notes attached directly to the pdf of the manuscript. However, in spite of al these shortcomings, the reviewed manuscript is ineresting and may be improved to make it fully fit for publication. Good luck with the revision!

Author Response

We sincerely appreciate the reviewer’s comments and suggestions. We have considered all aspects mentioned by him/her in the pdf document. Please, find attached our reviewed manuscript where we applied modifications paying special attention to each point mentioned.

We have rewritten our text to explain better the decisions related to each channel inspection, all data and statistical methods, conclusions to better tone our statements, and finally, some terms related to our research.

Yours sincerely.

Reviewer 3 Report

This is an interesting article. Though, the sample size was small, the results provide useful, insights for designers and manufacturers on customers perceptions on raw and recycled plastics. The authors should attend to the following to observations:

1. Highlight what will be covered in the article in the introduction.

2. The research method and research design have not be stated in Section 3: materials and methods. The methodology is indicated in Section 6: limitations and future research. This is too late and misplaced to indicate the study research method at the end of the paper.

3. Justify why the study opted for an experimental design approach?

4. How were the 36 participants sampled? Which sampling technique was used to select them. Justify your answer. 

5. Figures 5 to 7 can be placed at the appendices. Usually, questionnaires are not part of the manuscript.

6. Some references did not conform to the journal reference formatting, e.g., references no. 5, 10-14, 16, 22, 28-30.

Author Response

(The authors gave the same response as above.)

Round 2

Reviewer 1 Report

The manuscript is sufficiently improved and the authors have addressed satisfactory the reviewers' comments. Some minor language issues pesist. I would recommend a final proofreading before publishing. Issues spotted in the abstract: 

1. Line 14, “insight” or “view” please choose one word, both words are redundant.

2. Line 15, “natural” not “nature” raw materials. Please correct this throughout the manuscript.

Author Response

We sincerely appreciate the reviewer’s suggestions. We have considered all aspects mentioned by him/her. Please, find attached below our detailed response to the reviewer paying special attention to modifications done for each point mentioned. The response is done specifically for each comment as you suggested.  Finally, we would like to add that proofreading has been done to improve the readability of the paper.

Yours sincerely.

Reviewer 2 Report

The revised version of this paper needs still further revision. In particular, its linguistic side is unsatisfactory. These shortcomings do not lie only in very unsatisfactory English, but also in the fact that some statements are repeated in the same form throughout the text (for instance, lines 360-363 and 527-530; 432-434 and 438-440; 722-724 and 730-732). I indicated both these types of problems already in the review of the first version of this manuscript, but after the revision only a part of these shortcomings has beeen eliminated. Some amendments found in the revised version of this manuscript also produced less satisfactory effects than the respective statements found in its original version.

There are also other problems with this version of the manuscript. Among others, the authors provided a long explanation why they tested only 36 subjects, but they did not explain at all why they used only three materials. They also did not provide any exact data on statistical significance of effects they claim to discover thanks to EDA measurements. Therefore, all their conclusions related to these findings are not sufficiently supported by evidence. I indicated that problem already in the review of the first version of this manuscript, but it was left without any comments in its revised version.

The authors also did not explain how they quantified and analysed descriptive answers of their subjects found in their questionnaires (the subjects had to provide answers together with explanations why they answered in such a way). I indicated that problem too in the review of the first version of this manuscript, but it has not been dealt with by the authors as well.

Conclusions are much too long. The authors should provide conclusions more directly related to the results of their study.

Figure 16 contains two exactly the same graphs. One of them has to be removed.

I also put my detailed comments (in total123) directly in the pop-up notes attached to the pdf of the revised version of the reviewed manuscript.

Author Response

We sincerely appreciate the reviewer’s suggestions. We have considered all aspects mentioned by him/her. Please, find attached below our detailed response to the reviewer paying special attention to modifications done for each point mentioned. The response is done specifically for each comment as you suggested.  Finally, we would like to add that proofreading has been done to improve the readability of the paper.

Yours sincerely.

The revised version of this paper needs still further revision. In particular, its linguistic side is unsatisfactory. These shortcomings do not lie only in very unsatisfactory English, but also in the fact that some statements are repeated in the same form throughout the text (for instance, lines 360-363 and 527-530; 432-434 and 438-440; 722-724 and 730-732). I indicated both these types of problems already in the review of the first version of this manuscript, but after the revision only a part of these shortcomings has beeen eliminated. Some amendments found in the revised version of this manuscript also produced less satisfactory effects than the respective statements found in its original version.

The paper has been reviewed taking special attention to the sentences indicated. Finally, the repeated statements have been rewritten or eliminated.

There are also other problems with this version of the manuscript. Among others, the authors provided a long explanation why they tested only 36 subjects, but they did not explain at all why they used only three materials. They also did not provide any exact data on statistical significance of effects they claim to discover thanks to EDA measurements. Therefore, all their conclusions related to these findings are not sufficiently supported by evidence. I indicated that problem already in the review of the first version of this manuscript, but it was left without any comments in its revised version.

A new text has been added with information related to the use of only 3 materials in section 3.1.1. 

In order to improve and detail all the data analysis and statistical significance effect, we have now included a table that shows the significance for all fixed factors in the model (materials, channel, and the interaction between materials and channel). We keep the graphical output for the interpretation of results. We think this graphical output is very convenient as it facilitates interpretation once the analytical results show significance. With the graphs, all levels of categorical factors can be shown, thus avoiding the use of reference levels, which always make interpretation more complex.

The authors also did not explain how they quantified and analysed descriptive answers of their subjects found in their questionnaires (the subjects had to provide answers together with explanations why they answered in such a way). I indicated that problem too in the review of the first version of this manuscript, but it has not been dealt with by the authors as well.

Questions in the questionnaires that didn’t require giving a rating were studied in a qualitative way. The text has now been changed to include this information. No relevant conclusions were extracted from these descriptive questions.

Conclusions are much too long. The authors should provide conclusions more directly related to the results of their study.

Some sentences have been rewritten, but since there are a lot of topics we think it's quite relevant to comment and show all of them.

Figure 16 contains two exactly the same graphs. One of them has to be removed.

Thank you for the reminder, we have deleted one of them.

I also put my detailed comments (in total123) directly in the pop-up notes attached to the pdf of the revised version of the reviewed manuscript.

We have reviewed very carefully all comments to modify our manuscript and answer each of them properly.
